# Altered N6-Methyladenosine Modification Patterns and Transcript Profiles Contributes to Cognitive Dysfunction in High-Fat Induced Diabetic Mice

**DOI:** 10.3390/ijms25041990

**Published:** 2024-02-06

**Authors:** Zhaoming Cao, Yu An, Yanhui Lu

**Affiliations:** 1School of Nursing, Peking University, Beijing 100191, China; 2311110250@bjmu.edu.cn; 2Endocrinology Department, Beijing Chaoyang Hospital, Capital Medical University, Beijing 100020, China; anyu900222@126.com

**Keywords:** diabetic cognitive impairment, high-fat feeding, m^6^A methylation, epigenetic modification, hippocampal neuron

## Abstract

N6-methyladenosine (m^6^A) constitutes the paramount post-transcriptional modification within eukaryotic mRNA. This modification is subjected to stimulus-dependent regulation within the central nervous system of mammals, being influenced by sensory experiences, learning processes, and injuries. The patterns of m^6^A methylation within the hippocampal region of diabetes cognitive impairment (DCI) has not been investigated. A DCI model was established by feeding a high-fat diet to C57BL/6J mice. m^6^A and RNA sequencing was conducted to profile the m^6^A-tagged transcripts in the hippocampus. Methylated RNA immunoprecipitation with next-generation sequencing and RNA sequencing analyses yielded differentially m^6^A-modified and expressed genes in the hippocampus of DCI mice, which were enriched in pathways involving synaptic transmission and axonal guidance. Mechanistic analyses revealed a remarkable change in m^6^A modification levels through alteration of the mRNA expression of m^6^A methyltransferases (METTL3 and METTL14) and demethylase (FTO) in the hippocampus of DCI mice. We identified a co-mediated specific RNA regulatory strategy that broadens the epigenetic regulatory mechanism of RNA-induced neurodegenerative disorders associated with metabolic and endocrine diseases.

## 1. Introduction

Patients with type 2 diabetes mellitus (T2DM) are more likely to develop mild cognitive impairment (MCI) than those without T2DM, and patients with T2DM and MCI are more likely to progress to Alzheimer’s disease (AD) than those without either disease [1]. Epidemiological data show that patients with T2DM have a 1.4- to 1.8-fold higher risk of MCI and a 1.5- to 2.5-fold higher risk of dementia than the general population [2]. Cognitive impairment due to hippocampal damage in patients with T2DM mainly includes abnormalities of memory function, attention, visual–spatial cognitive function, executive function, and information processing speed [3]. Diabetic cognitive impairment (DCI), an irreversible degenerative disease of the central nervous system, has remained a major problem because the exact mechanism underlying its development has not been fully elucidated and effective means of prevention and treatment are lacking [4]. DCI burdens the patient’s family and society [5]. An understanding of the precise pathogenic mechanisms of DCI could lead to strategies for its early detection, identification, and intervention, which would effectively prolong the onset of hippocampal damage in individuals with T2DM, thereby, averting the progression of dementia and AD and ultimately enhancing the overall quality of life of patients.

N6-methyladenosine (m^6^A) is an RNA modification that affects the splicing, export, translation, and degeneration of mRNAs [6]. As a dynamic modification, m^6^A plays vital roles in pathological and physiological processes in eukaryotes [7]. Post-transcriptional methylation is performed by three types of enzymes: methyltransferase complexes (writers), demethylases (erasers), and binding proteins (readers) [8]. Previous studies have indicated a strong association between m^6^A modifications and metabolic disorders, including obesity and T2DM [9]. Studies have also revealed an association between m^6^A and neurodegenerative diseases, including glioblastomas [10], traumatic hippocampal injury [11], and Parkinson’s disease [12]. m^6^A is highly abundant in the mammalian hippocampus [13] and plays important regulatory roles in synaptic function [14], axonal regeneration [15], neuronal development, and neurogenesis [16]. m^6^A-seq of mouse hippocampal synaptosomal mRNAs revealed extensive distribution of m^6^A in the distal pre- and postsynaptic regions. Moreover, functional enrichment analysis showed that 1266 hyper-methyl esterified genes were enriched in synapse-related functional pathways, including synaptic maturation, organisation, assembly, and regulation of synaptic transmission [17]. m^6^A is dynamically regulated during learning. Learning training enhances m^6^A modifications in immediate early genes related to memory, such as ARC and CFO. Additionally, this training facilitates the translation of these genes, indicating a strong correlation between m^6^A mRNA modification and learning and memory formation processes [18]. m^6^A also promotes hippocampus-dependent learning and memory through a binding protein encoded by YTHDF1 [19]. These findings improve our understanding of the mechanisms underlying the regulation of learning and memory at the RNA level. Moreover, they provide a new direction for studying the molecular mechanisms underlying damage to hippocampal neurons.

This study aimed to establish the profile of m^6^A modifications in mice with DCI and elucidate the potential regulatory mechanisms of m^6^A methylation underlying DCI. We performed methylated RNA immunoprecipitation with next-generation sequencing (MeRIP-seq) and RNA-Sequencing (RNA-seq) to analyse the differences in gene methylation and mRNA expression in mice with DCI and verified the change in the expression of methylase and its regulatory role in DCI. In this study, we identified genes that were differentially m^6^A-modified and expressed in the DCI mouse hippocampus. These genes were enriched in pathways related to synaptic transmission and axonal guidance. Our results show that abnormalities in methylation are the genetic mechanisms underlying the development of DCI.

## 2. Results

### 2.1. Changes in Body Weight and Fasting Blood Glucose Levels between the Two Groups of Mice

At the start of the experiment, the mean body weight of mice in the DCI (19.36 ± 0.24 g) and control (19.18 ± 0.15 g) groups was not statistically significant (*p* = 0.511). Similarly, at the start of the experiment, the mean blood glucose level in the DCI (4.20 ± 0.69 mmol/L) and control (4.12 ± 0.60 mmol/L) groups was not statistically significant (*p* = 0.801). The bodyweight of mice in both groups increased as the experiment progressed, and the differences between the groups were statistically significant one week after the start of the experiment (*p* < 0.01; Figure 1A). One week after the start of the experiment, both groups had elevated fasting blood glucose, and it was significantly higher in the DCI group than in the control group (*p* < 0.01; Figure 1B). During the experiment, the hyperglycaemic state of mice in the DCI group persisted, suggesting that the T2DM mouse model was successfully established.

### 2.2. Results of the Morris Water Maze Test

In the Morris water maze test (Figure 2), all mice showed shorter escape latency on day 5 than on day 1. The difference was not statistically significant on day 1 (*p* = 0.985), but the escape latency of mice in the control group was significantly shorter than that of mice in the DCI group on days 2–4 (Figure 2A). The difference in swimming speed between the two groups was not statistically significant (Figure 2B). The average swimming distance for mice in the DCI group was significantly higher on day 5 than that for mice in the control group (Figure 2C). The time spent in the target quadrant (*p* < 0.05, Figure 2D) and the number of platform crossings (*p* < 0.01, Figure 2E) for mice in the DCI group were significantly lower than for mice in the control group, indicating the development of cognitive impairment and the successful establishment of the DCI mouse model.

### 2.3. H&E Staining of the Hippocampus in the Two Groups of Mice

Changes in the hippocampus of DCI mice were assessed using H&E staining. As shown in Figure 3, the hippocampal cells in the control group were tightly packed and showed a regular morphology and uniform staining. In contrast, in the DCI group, the number of neurons was reduced, and they were sparsely arranged and had condensed nuclei.

### 2.4. GO and KEGG Pathway Analyses of Differentially Expressed Genes (DEGs) in the Hippocampus of DCI Mice

To explore the differences in transcriptome profiles, we performed RNA-seq analysis of hippocampal samples obtained from control (*n* = 3) and DCI (*n* = 3) mice. The expression levels of 666 genes were significantly altered, of which 297 were upregulated and 369 were downregulated (Figure 4A). Table 1 shows the top 20 DEGs in the hippocampus of DCI mice. GO analysis showed that the upregulated genes were most enriched in nucleobase-containing compound, cellular nitrogen compound, heterocycle, and cellular aromatic compound metabolic processes (Figure 4B). In contrast, downregulated genes were enriched in several metabolic pathways, cellular component organization, cellular amide metabolic process, and ganglioside metabolic process (Figure 4C). KEGG pathway analysis further revealed that the upregulated genes were enriched in RNA transport, spliceosome, and mRNA surveillance pathways (Figure 4D). The downregulated genes were associated with metabolic pathways, including those implicated in non-alcoholic fatty liver disease, Parkinson’s disease, and Alzheimer’s disease (Figure 4E).

### 2.5. Altered m^6^A Modification of Genes in the Hippocampus of DCI Mice

To compare the m^6^A modification of genes between the control (*n* = 3) and DCI (*n* = 3) groups, we performed a transcriptome-wide m^6^A-seq analysis using MeRIP-seq. In total, 23,711 non-overlapping m^6^A peaks in the control group were found within 10,664 coding gene transcripts (mRNAs) in the DCI group. In contrast, the DCI group exhibited 23,751 non-overlapping m^6^A peaks across 10,809 distinct mRNAs in the three replicates of the control group. Of these, 9345 methylated genes overlapped between the control and DCI groups (Figure 5). A total of 1278 differentially methylated m^6^A sites (DMMSs) were identified, of which 60.88% (778/1278) and 39.12% (500/1278) showed significantly higher and lower levels of methylation, respectively, in DCI versus the control. Table 2 shows the top 10 up- and down-methylated m^6^A sites with the highest fold-change values (DCI vs. control). To analyse their distributional profiles, m^6^A peaks were distributed throughout the RNA (Figure 6A) and were divided into the following five groups based on the transcript regions in which they were distributed: 5′UTR, 3′UTR, segments of the transcription start site region, stop codon segments, start codon segments, and coding sequences (CDS). The distribution of m^6^A peaks was significantly enriched in the vicinity of the 3′UTR, CDS, and stop codons (Figure 6B). To assess the regions of m^6^A modification on these genes, those annotated by peak were screened, and the coverage of reads on these genes were determined. In the DCI group, m^6^A modification regions were biased toward the 3′UTR region (Figure 6C). All the DMMSs within mRNAs were mapped to chromosomes to assess their distribution profiles. The m^6^A modification profiles differed across mouse chromosomes.

To examine the biological function of m^6^A modification in DCI mice, protein-coding genes containing DMMSs from the two groups were selected for GO and KEGG analyses (Figure 7A). In the GO analysis, the hyper-methylated genes were significantly enriched in the cellular process, regulation of biological process, catalytic activity, and heterocyclic compound binding. The KEGG pathway analysis revealed that hyper-methylated genes were associated with the T-cell receptor signalling pathway, small cell lung cancer, and RIG-I-like receptor signalling pathway (Figure 7B). For the hypo-methylated genes, GO analysis revealed significant enrichment in the cellular process, metabolic process, binding organic cyclic compound, binding heterocyclic compound, and catalytic activity (Figure 7C). The KEGG pathway analysis revealed the association of hypo-methylated genes with hypertrophic cardiomyopathy and the Hippo signalling pathway (Figure 7D).

### 2.6. Differential m^6^A-Modification and Expression of Genes in the Cerebral Cortex Result from Altered m6A Methyltransferase and Demethylase Levels

To investigate how DCI altered the m^6^A modification and affected gene expression, we evaluated the expression of 26 m^6^A RNA methylation regulators selected from the published literature; these included nine writers (METTL3, METTL14, METTL16, WTAP, VIRMA, ZC3H13, RBM15, RBM15B, CBLL1), 15 readers (YTHDC1, YTHDC2, YTHDF1, Ythdf2, YTHDF3, HNRNPC, FMR1, LRPPRC, HNRNPA2B1, IGFBP1, IGFBP2, IGFBP3, RBMX, ELAVL1, IGF2BP1), and two erasers (FTO and ALKBH5). Compared with the control group, the expression of METTL14 and FTO was significantly downregulated in the DCI group. The expression of METTL3 was significantly upregulated in the DCI group compared with that in the control group (*p* < 0.05) (Figure 8B).

To determine the alterations in m^6^A modification-related protein levels in the hippocampal tissues of diabetic mice, we measured the mRNA and protein levels of METTL3, METTL14, and FTO in the hippocampal tissue of mice using quantitative real time polymerase chain reaction (qRT-PCR) and western blotting. As shown in Figure 8C,D, compared with the control group, the mRNA and protein levels of METTL3 were increased, and those of METTL14 and FTO were decreased in the hippocampal tissue of diabetic mice, which was consistent with the results of RNA sequencing analysis.

### 2.7. Conjoint Analysis for m^6^A MeRIP-Seq and RNA-Seq Data

The combined analysis of DMMSs and DEGs yielded 163 mRNAs with significantly altered m^6^A peaks and mRNA levels. Both m^6^A and mRNA levels were upregulated for 41 mRNAs and downregulated for 14 mRNAs. Moreover, 30 genes showed upregulated mRNA expression and downregulated m^6^A levels, whereas 78 genes showed downregulated mRNA expression and upregulated m^6^A levels (Figure 9A). Finally, a PPI network was constructed to show the connections between the proteins encoded by the 163 genes (Figure 9B). GO and KEGG analyses for four groups ((1). m^6^A upregulated and mRNA downregulated; (2). m^6^A upregulated and mRNA upregulated; (3). m^6^A downregulated and mRNA downregulated; and (4). m^6^A downregulated and mRNA upregulated) of genes are shown in Figure 10. In the GO analysis, the m^6^A upregulated and mRNA downregulated genes were significantly enriched in the cellular metabolic process, aromatic compound biosynthetic process, and regulation of RNA metabolic process. The KEGG analysis showed that these genes were enriched in fatty acid metabolism, small cell lung cancer, propanoate metabolism, and pyruvate metabolism. For the m^6^A upregulated and mRNA upregulated genes, GO analysis revealed enrichment in ferrous iron binding, sulfinoalanine decarboxylase activity, aspartate 1-decarboxylase activity, and negative regulation of cell junction assembly and KEGG analysis revealed enrichment in the type 2 diabetes mellitus, adipocytokine signalling, and insulin signalling pathways. The m^6^A downregulated and mRNA downregulated genes were enriched in oligopeptide binding, glutathione binding, and disulphide oxidoreductase activity in the GO analysis, whereas they were enriched in the citrate cycle (TCA cycle), propanoate metabolism, and apoptosis—multiple species in the KEGG analysis. The m^6^A downregulated and mRNA upregulated genes were enriched in embryonic skeletal joint morphogenesis, activation of protein kinase activity, and NADP metabolic process in the GO analysis and in neomycin, kanamycin, and neurotrophin signalling pathway in the KEGG analysis.

## 3. Discussion

Epidemiological and animal studies have shown that T2DM increases the risk of developing neurodegenerative diseases [20]. However, the molecular mechanisms through which T2DM affects the central nervous system remain unknown. m6A, the most common post-transcriptional modification, plays an essential role in several biological processes. In vivo, the hippocampus exhibits a developmentally altered abundance of m^6^A methylation. m^6^A methylation is biased toward neuronal transcripts and is sensitive to neuronal activity [21]. In the hippocampus, m^6^A regulates several developmental and physiological processes, including neurogenesis, axonal growth, synaptic plasticity, circadian rhythms, cognitive function, and stress response [22]. With continuous progress in high-throughput sequencing technology, increasing attention to the m^6^A landscape has been focused on neoplastic and non-neoplastic diseases of the CNS. Revealing the m^6^A landscape in the hippocampus of DCI mice will help explore new mechanisms underlying DCI and should provide new targets for prevention and treatment. In this study, we elucidated the epigenetic mechanisms of m6A in the hippocampus of DCI mice. First, we found that DM induction in mice fed a high-fat diet caused cognitive disorders and damage to hippocampal neurons. Next, we performed MeRIP-seq and RNA-seq and identified 1278 m^6^A peaks to be significantly differentially methylated. Among them, 500 peaks were significantly downregulated in DCI mice. Employing the conjoint MeRIP-seq and RNA-seq analyses, we determined the presence of DEGs and differentially modified RNAs in the control and DCI groups. Based on our findings, we speculate that alterations in m^6^A modifications in the hippocampus may be responsible for diabetic nerve damage, which may lead to the development of cognitive disorders in patients with diabetes.

Cognitive function-related hippocampal structures, such as the hippocampus, are susceptible to hyper- and hypo-glycaemia. A vast spectrum of pathological changes has been observed in rodent models of diabetes, including synaptic alterations, decreased cell proliferation, increased microvascular permeability, neuron loss, and hippocampal atrophy [23]. In our study, behavioural tests showed that DCI mice exhibited memory impairment. In addition, we found that the DCI mouse model suffered from structural damage to the hippocampus, as evidenced by the results of H&E staining. Next, RNA-seq analysis revealed that the genes with upregulated mRNA levels (compared with the control group) were significantly enriched in RNA transport, spliceosome, mRNA surveillance pathway, and RNA degradation. The genes with downregulated mRNA levels were significantly enriched in non-alcoholic fatty liver disease, Parkinson’s disease, Alzheimer’s disease, and metabolic pathways. MeRIP-seq analysis revealed that methylation in the hippocampus tissue of DCI mice was characterised by more hyper-methylated regions on chromosomes 1, 7, 9, and 12 compared with that in the control, and these hyper-methylated regions were associated with neurodegenerative diseases and specific immune-related pathways, axonal guidance, NF-kappaB signalling pathway, T-cell receptor signalling pathway, and FoxO signalling pathway. More hypo-methylated regions were observed on chromosomes 2, 3, 5, 11, 15, and X, and these downregulated methylation genes were enriched in pathways associated with T2DM and cell signalling, such as transcriptional misregulation in cancer, hypertrophic cardiomyopathy, Hippo signalling pathway, mTOR signalling pathway, basal cell carcinoma, longevity regulating pathways-multiple species, FoxO signalling pathway, Toll-like receptor signalling pathway, signalling pathways regulating pluripotency of stem cells, and TNF. Taken together, genes associated with neurodevelopmental and neurodegenerative changes were characterised by the upregulation of m^6^A methylation and downregulation of mRNA expression in the hippocampus of DCI mice. In contrast, genes associated with T2DM, obesity, and cancer showed a trend toward downregulation of m6A methylation and upregulation of mRNA expression.

To investigate the mechanism by which DCI alters m^6^A modification and affects gene expression, we evaluated 26 previously reported regulators of m^6^A RNA methylation. We found that m^6^A methylase, METTL3, METTL14, and FTO were significantly altered in the DCI group compared with that in the control group. METTL3 and METTL14 are the essential subunits of the m^6^A writer complex. METTL3 is the catalytic subunit and METTL14 activates METTL3 via allostery and recognition of RNA substrates [24]. Many investigators have evaluated the necessity of m^6^A in neuronal function by conditionally deleting METTL14. This deletion reduced striatal m^6^A levels without altering the number or morphology of cells, and significantly reduced the total m^6^A level in mRNAs isolated from the substantia nigra region, thereby, altering dopaminergic neuron function [25]. In the present study, the expression of METTL14 was significantly downregulated in the DCI group, which may have led to the inability to form a stable METTL3 and METTL14 complex conformation and reduced the catalytic activity. Reduction in m^6^A levels can induce apoptosis in dopaminergic neurons by elevating oxidative stress and Ca^2+^ influx [26]. FTO was the first demethylase involved in m^6^A to be discovered [27]. FTO mediates m^6^A and m1A RNA demethylation [19]. m^6^A-dependent demethylation by FTO is associated with dopaminergic neurotransmission, adult neurogenesis, and axonal elongation. Moreover, elevated FTO levels crucially influence glucose and lipid metabolism-related gene expression in patients with DM [28]. In the present study, the FTO expression was reduced in the hippocampus of DCI mice, consistent with previous findings [29]. Therefore, we speculate that high fat diet-induced T2DM is secondary to FTO downregulation, which is a protective mechanism in organisms; however, the role of FTO in the CNS of patients with diabetes remains to be investigated.

To better understand the roles of m^6^A methylation in T2DM patients with cognitive impairment, we screened all the differentially expressed peaks combined with the differentially expressed mRNA ((1). m^6^A upregulated and mRNA downregulated; (2). m^6^A upregulated and mRNA upregulated; (3). m^6^A downregulated and mRNA downregulated; and (4). m^6^A downregulated and mRNA upregulated). The m^6^A upregulated mRNA-downregulated genes include Pcgf2, Map6d1, Angpt2, Rec8, and Dcaf5. Most of these genes, such as transcription factor family genes Mybl1 [30] and Fnip2, are related to the occurrence and development of T2DM and morbid obesity [31]. Wnt10a, which is associated with both Parkinson’s disease and fat accumulation, exhibits a direct correlation with the inflammatory cytokine gene Lin37. Although Wnt10 and Lin37 are not directly related to the occurrence of diabetes and morbid obesity, the results of the PPI network analysis showed that these genes directly interact with genes associated with the development of T2DM, such as Mybl1. KEGG enrichment analysis showed that these genes were significantly enriched in the mTOR signalling pathway, metabolic pathway, fatty acid metabolism, and mTORComplel1-S6K1 signalling pathway, which is directly related to insulin resistance [32]. The genes with upregulated m^6^A peak and upregulated mRNA included Ddx5, Wdr89, Nrarp, Lyrm9, Stum, and Ttc14. The enrichment analysis revealed that most of these genes were related to immune response. Among these, Irs4 is directly associated with insulin resistance [33]. KEGG enrichment analysis showed that these genes were significantly enriched in T2DM, FoxO signalling pathway, and PI3K-Akt signalling pathway. Activation of the PI3K/AKT pathway can accelerate apoptosis, and most studies have confirmed the inflammatory cascade reactions. The inflammatory cascade can downregulate glucose transporters and insulin-related molecules (such as GLUT4, cyclin A, cyclin E, and IRS-1), leading to elevated blood glucose and insulin resistance in patients with T2DM [34]. Genes with downregulated m6A peaks and downregulated mRNAs levels included Gins2, Efs, Emd, Suclg1, Entpd3, Gpbp1l1, Ngfr, and Pfkfb3. The enrichment analysis revealed that most of these genes were related to metabolism and cell cycle. Among these, the cyclic nucleic acid replication factor Gins2 is an important component of the replication fork uncapping enzyme GINS complex in cell cycle [35]. In addition, mice knocked out for Ngfr exhibited a delayed initial growth rate and a neurological manifestation of the hind-limb grip response in the tail suspension test [36]. The PPI network analysis showed that Ngfr directly interacts with genes associated with neurodegenerative diseases, such as Bmp4, Ngf, and Angpt2. Therefore, Ngfr may indirectly contribute to the development of neurodegenerative diseases by acting on these genes. The genes with downregulated m^6^A peak and upregulated mRNA levels included Mapk13, Adamts19, Bpifb9b, Lrrc71, Dnali1, Cd320, Sp7, Mfap2, and Ngf. Pathway enrichment analysis has shown that these genes are significantly enriched in neurodegenerative pathologies, such as Parkinson’s disease, Alzheimer’s disease, and depression. For example, it has been shown that Cd320 is a cell membrane surface receptor involved in the intracellular transport of vitamin B12, which is integral to the function of the nervous system and is involved in lipid synthesis and maintenance of metabolism and function of nerve myelin [37]. Other genes in this group are involved in various neurodegenerative diseases through their involvement in apoptosis, oxidative stress, and neurotransmitter imbalance [38].

Neurodegenerative diseases are caused by many factors, including oxidative stress, neurotransmitter imbalance, genetic and epigenetic factors, and defects in neurogenesis [37]. Our findings in this study indicate that genes exhibiting a decrease in m^6^A modification but an increase in mRNA expression within the hippocampus of mice fed a high fat diet are implicated in the pathogenesis of diseases, such as T2DM and obesity. A strong trend towards the downregulation of genes related to neurodevelopment, including synaptic transmission and CNS developmental pathways, and cell signalling, was found, in addition to altered mRNA expression of methylation-related enzymes. The PPI network analysis revealed significant association between genes implicated in neurodegenerative disorders and those involved in the progression of T2DM, suggesting a direct interaction and mutual influence between them.

## 4. Materials and Methods

### 4.1. Animals

Specific pathogen-free (SPF) male C57BL/6J mice (6-weeks-old, 19.08 ± 0.58 g) were purchased from the Beijing Weitong Lihua Experimental Animal Technology Co., Ltd. [Experimental Animal License number: SCXK (Beijing, China) 2019-0010]. All mice were raised in an animal facility, at an indoor temperature of 18–22 °C and relative humidity of 40–60%, under 12 h alternating dark/light cycles. All the mice were allowed free access to food and water. Before starting the experiments, all animals were housed in an animal house environment for one week. The animals were randomly divided into the DCI (*n* = 9) and control groups (*n* = 9), with nine animals in each group. The DCI group was fed a high-fat diet (Cat #D12492, Research Diets, USA; 60 kcal% fat) for six weeks, whereas the control group was fed a standard-fat diet (SFD; Cat #1022, Beijing HFK Bioscience, China; 10 kcal% fat). The study protocol was approved by the Animal Ethics Committee of the University (approval Number: LA2019184).

### 4.2. Behavioural Test

After establishing the success of the T2DM animal model, we immediately carried out the water maze experiment (week 7, 5 days). The Morris water maze (MWM) test is commonly used to evaluate spatial learning and memory in rodents. The experimental apparatus comprised a black circular water pool (diameter, 1.50 m; height, 0.60 m). The water temperature in the pool was maintained at 24 ± 2 °C. The pool had a featureless inner surface divided into four equal quadrants (A–D) [21]. A translucent 10 × 10 cm platform, submerged 1 cm below the water surface, was hidden in the centre of quadrant NE (target quadrant) during the training period and removed at the time of the probe task. Training was conducted three times daily for five consecutive days before the probe task. Each mouse was allowed to swim until it found the platform or until 60 s had elapsed. The mice were then left on the platform for 10 s. The platform was removed from the pool for the space exploration task and the mice were allowed to swim for 60 s. The swim escape latency, path length, and time spent in the target quadrants were recorded using a video tracking system (Zhongshi Dichuang Information Technology Co., Ltd., Beijing, China).

### 4.3. Tissue Sectioning and Staining

Researchers randomly selected three mice from each of the two groups for haematoxylin and eosin (H&E) staining. H&E staining was performed according to the following procedure: the hippocampus tissue was quickly excised and immersed in 4% paraformaldehyde at 4 °C for 24 h before being embedded in paraffin. A tissue slicer was used to obtain coronal sections (4 µm thick). Finally, the sections were de-waxed with xylene and dehydrated using an ethanol gradient before H&E staining, following the manufacturer’s instructions (Beyotime, Shanghai, China) to visualise structural damage.

### 4.4. High-Throughput m^6^A-Seq and RNA-Seq

The mice were anaesthetised and transcardially perfused with 200 mL of saline at 4 °C. The hippocampus was then carefully dissected and stored in liquid nitrogen. Total RNA from the hippocampus was extracted after lysing it with Tissuelyser-24 (Shanghai, China) using the TRIzol™ reagent. Total RNA (20 µg) was used, and three biological replicates were used for the control and DCI groups. The RNA samples were chemically fragmented to obtain 200 nt fragments. The fragmented RNA was incubated overnight with an anti-m^6^A antibody and Dynabeads. After washing three times with 1× IP buffer and three times with 1× wash buffer, the immunoprecipitated mRNA fragments were extracted with phenol-chloroform and precipitated with ethanol. m6A antibody-enriched mRNA and input mRNA libraries were constructed as follows: cDNA was synthesised from total RNA after removing the rRNA, followed by PCR amplification; finally, the completed libraries were purified on a BiOptic Qsep100 Bio-Fragment Analyzer (Bioptic, Changzhou, Jiangsu, China) for quality control. The libraries were sequenced on an Illumina NovaSeq platform following the PE150 protocol.

### 4.5. Sequencing Data Processing

Adapters and filter sequences were trimmed using Cutadapt (v2.5.0). Sequencing data quality, mass distribution, base content distribution, and repeated sequencing fragment proportion were analysed using FastQC version 0.11.5 [39]. The remaining reads were aligned to the human ensemble genome, GRCh38 (mouse ensemble genome GRCm38), using the Hisat2 aligner (v2.1.0) with the following parameter: “-rna-strandedness RF”. m^6^A peaks were identified using the exomePeak R package (v2.13.2) with the following parameters: “PEAK_CUTOFF_PVALUE 0.05, PEAK_CUTOFF_FDR NA, FRAGMENT_LENGTH 200”. The m^6^A peaks with a *p*-value < 0.05 were chosen for subsequent de novo motif analysis using homer (v4.10.4) with the following parameter: “-len 6-RNA”. m^6^A-RNA-related genomic features were visualised using the Guitar R package (v1.16.0). The HOMER software version 4.11 was used to analyse the motifs of the m^6^A peaks [40]. The BAM files of sequencing results were visualised using IGV version 2.11.9 [41].

### 4.6. Identification of DEGs and DMMSs

The DiffReps software version 1.55.3 was used to identify the DEGs. The default filter criteria were *p*-value < 0.05 and |fold change| > 0.5. Screening of differential m6A peaks was conducted using the exomePeak R package, and the filtering threshold was set at *p*-value < 0.05 and |fold change| > 1.

### 4.7. Kyoto Encyclopedia of Genes and Genomes (KEGG) and Gene Ontology (GO) Analysis of DEGs and DMMSs

The GO program (http://www.geneontology.org) comprises a structured controlled vocabulary of annotated genes, gene products, and sequences. We performed GO functional analysis to annotate and speculate on the potential roles of the DEGs and DMMSs. The GO terms with *p*-value ≤ 0.05 were regarded to be significantly enriched. The KEGG pathway analysis coordinates the molecular datasets of metabolomics, transcriptomics, genomics, and proteomics onto a KEGG pathway map to explain the biological functions of these molecules. A KEGG term with a *p*-value ≤ 0.05 was considered significantly enriched.

### 4.8. Protein–Protein Interaction (PPI) Network Analysis

We conducted a conjoint analysis of genes showing differential expression and differential m^6^A modification and used the *p*-value and fold change criteria to screen candidates for PPI network analysis. These DEGs were imported into the STRING database, which contains comprehensive information about interactions between proteins, to determine the interactions between genes [42]. The PPI network was constructed by importing data into the Cytoscape 3.5.1 software, and the network was analysed using a network analyser. Genes showing interactions with combined scores greater than 0.4 were selected to construct a PPI network diagram [43].

### 4.9. Validation of Gene Expression Levels

After evaluating the gene expression of 26 m^6^A RNA methylation regulators screened from the published literature, we found significant differences in the expression of METTL3, METTL14, and FTO between the two groups of mice. The expression levels of METTL3, METTL14, and FTO were evaluated in hippocampal tissue samples from mice in the control and DCI groups. Total RNA was extracted from frozen tissue samples using TRIzol reagent (Waltham, MA, USA), followed by centrifugation and isopropanol precipitation. The RNA pellets were washed with ethanol, air-dried, and dissolved in RNase-free water. Quantitative reverse transcription polymerase chain reaction (qRT-PCR) was performed using diluted cDNA products, with ACTB as an internal control. The primers used for qRT-PCR were designed using the Primer3 website, Primers used in this study shown in Appendix A. The protein levels of METTL3, METTL14, and FTO were quantified using western blot analysis. The hippocampal tissue was homogenised in RIPA buffer and the supernatant was collected. The protein content in the supernatant was quantified using the BCAN Protein Assay Kit (Waltham, MA, USA). Equal quantities of total protein extracts were electrophoresed, and the proteins were transferred onto PVDF membranes. After blocking and incubation with primary antibodies, the membranes were incubated with corresponding secondary antibodies. The membranes were then washed, and the protein bands were visualised using an enhanced chemiluminescence kit. The expression levels of proteins were determined by densitometry analysis using the ImageJ version 1.8.0.

### 4.10. Statistical Analysis

All statistical analyses were conducted using the R software (version 4.2.2). The values are presented as mean ± standard deviation (SD) for continuous numerical data with normal distribution. Normal distribution test using the Shapiro–Wilk test. Homogeneity of variance test using the Bartlett test. Student’s *t*-test was used to compare two groups of normally distributed data with equal variance, and the Welch’s *t*-test was employed for normally distributed data with unequal variance. A *p*-value of <0.05 was considered statistically significant.

## 5. Conclusions

This study demonstrates that the perturbation of glucose metabolism resulting from feeding a high fat diet induces changes in the expression of enzymes related to m^6^A modification. These alterations lead to modifications in the methylation patterns of genes associated with neurodegenerative disorders, particularly in the hippocampal region of mice. These findings provide new insights into the mechanism of high fat diet-induced cognitive dysfunction in diabetic mice, which could lead to a new therapeutic strategy for diabetes-induced hippocampal lesions. Further studies should prioritise the validation of these genes to ascertain the precise pathways associated with this pathological process. This validation should be conducted through in vitro and in vivo experiments, wherein methylation-related enzymes are manipulated to observe their effect on gene expression, the diabetic hippocampus is analysed for apoptosis, and its correlation with cognitive status is elucidated.

## Figures and Tables

**Figure 1 ijms-25-01990-f001:**
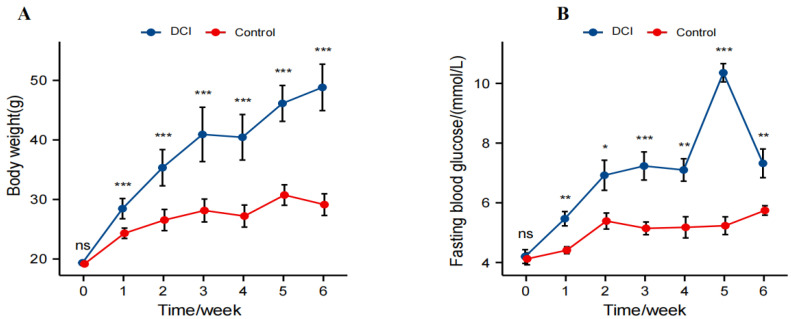
Comparison of changes in body weight and fasting blood glucose between two groups of mice before and after modelling (**A**) bodyweight; (**B**) fasting blood glucose. Significance identification: Not significant; ns, *p* ≥ 0.05; *, *p* < 0.05; **, *p* < 0.01; ***, *p* < 0.001.

**Figure 2 ijms-25-01990-f002:**
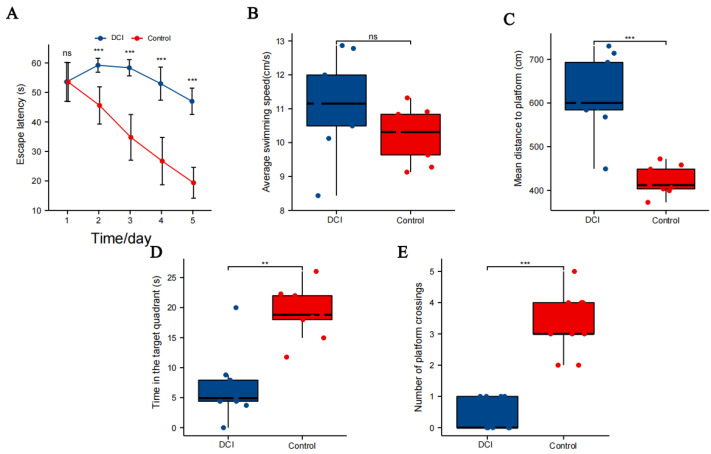
Comparison of spatial probe test between the two groups (*n* = 9 for each group) of mice in the Morris water maze. (**A**) Distance to the platform; (**B**) average swimming speed the orientation navigation trial; (**C**) mean distance to platform; (**D**) time in the target quadrant; (**E**) number of crossing platforms. Significance identification: Not significant; ns, *p* ≥ 0.05; **, *p* < 0.01; ***, *p* < 0.001.

**Figure 3 ijms-25-01990-f003:**
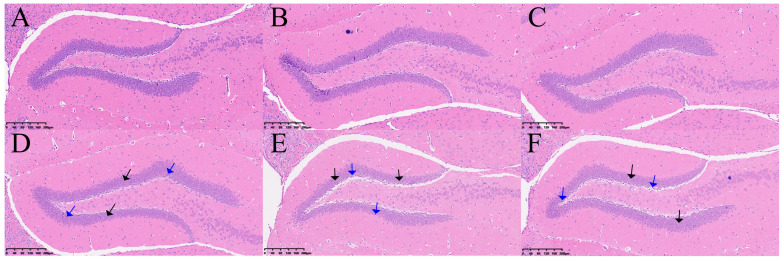
H&E staining results in the hippocampus of two groups of the mice. Control group (*n* = 3): (**A**–**C**), DCI group (*n* = 3): (**D**–**F**). Black arrow: sparse arrangement and reduced number of neuronal cells; blue arrow: neuronal nuclei solidified.

**Figure 4 ijms-25-01990-f004:**
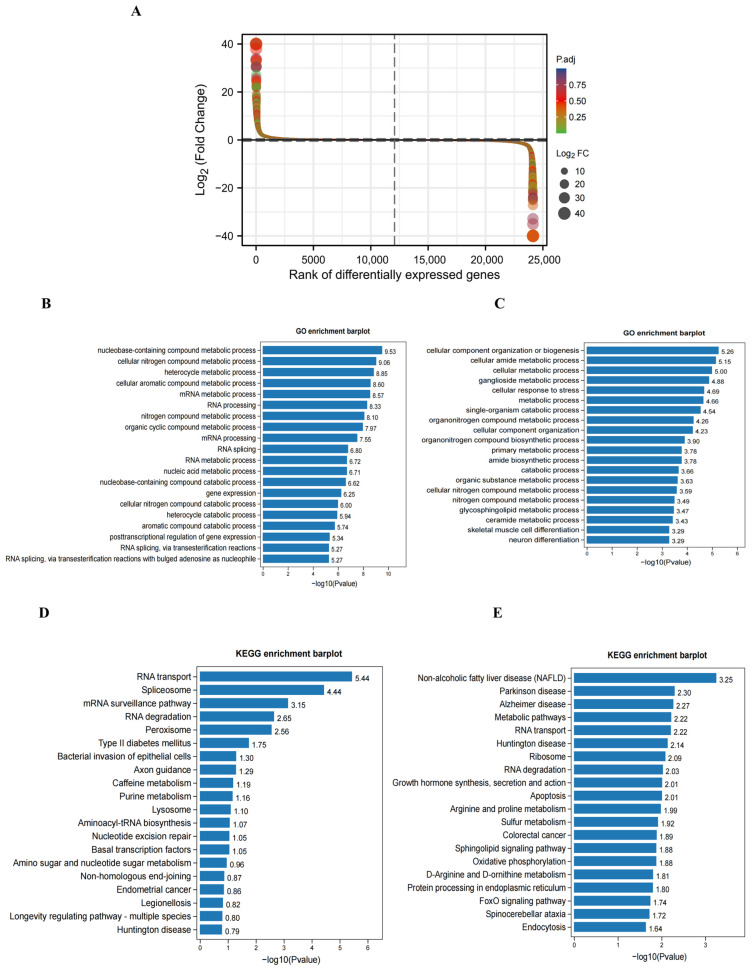
Differential expression of genes in cerebral cortex between control (*n* = 3) and DCl (*n* = 3) mice. (**A**) Difference ranking of RNA sequencing result. (**B**) The top twenty items of the most enriched GO analysis of upregulated genes. (**C**) The top twenty items of the most enriched GO analysis of downregulated genes. (**D**) The enriched KEGG enrichment analysis of upregulated genes. (**E**) The enriched KEGG enrichment analysis of downregulated genes.

**Figure 5 ijms-25-01990-f005:**
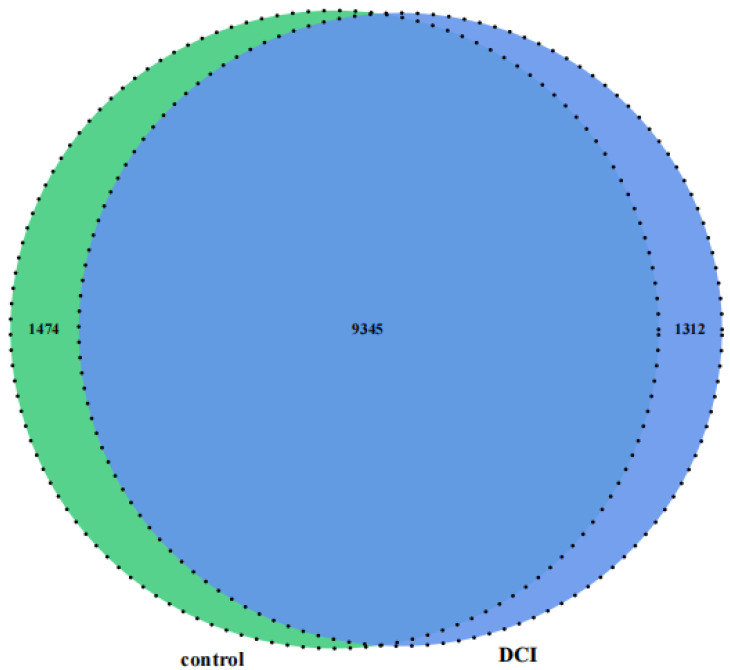
Venn diagram of m^6^A-modified genes in the control group and the DCl group.

**Figure 6 ijms-25-01990-f006:**
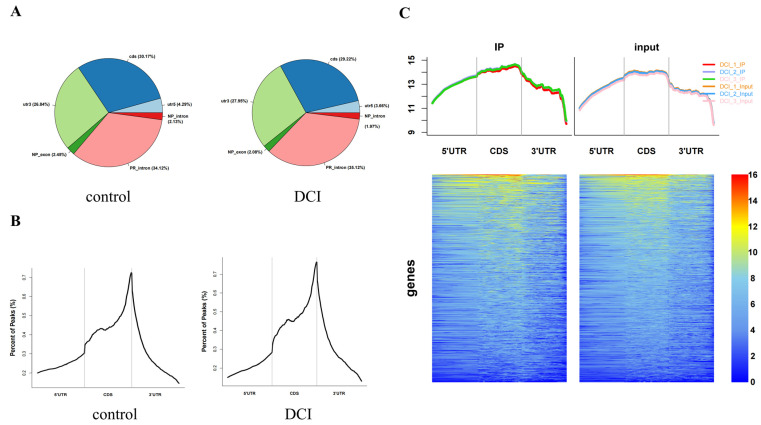
Overview of altered m^6^A methylation map in the hippocampus of mice. (**A**). The distribution of peaks in different regions of the gene. CDs are the CDS region of the gene, utr5 andutr3 are the 5′UTR and 3′UTR of the gene, respectively, PR intron is the intronic region of the coding gene, and NP_exon and NP_intron are the exons and intronic regions of the non-coding gene. (**B**). Distribution of peak in the different areas on gene exons. The horizontal coordinates are the 5′UTR, CDS, and 3′UTR, and the vertical coordinates are the relative proportions of the distribution of different peak regions. (**C**). Heat map of reads distribution of peak-associated genes in the DCI group. The left and right graphs represent the distribution of reads in IP and input samples on the functional regions of the genes annotated to peak, respectively. The top graph shows the cumulative distribution of reads over all functional regions of genes (Total reads are taken as a logarithm of 10), and the bottom graph shows the distribution of reads on each gene, with the colour gradient from blue to yellow to red, to represent the depth of coverage from light to dark.

**Figure 7 ijms-25-01990-f007:**
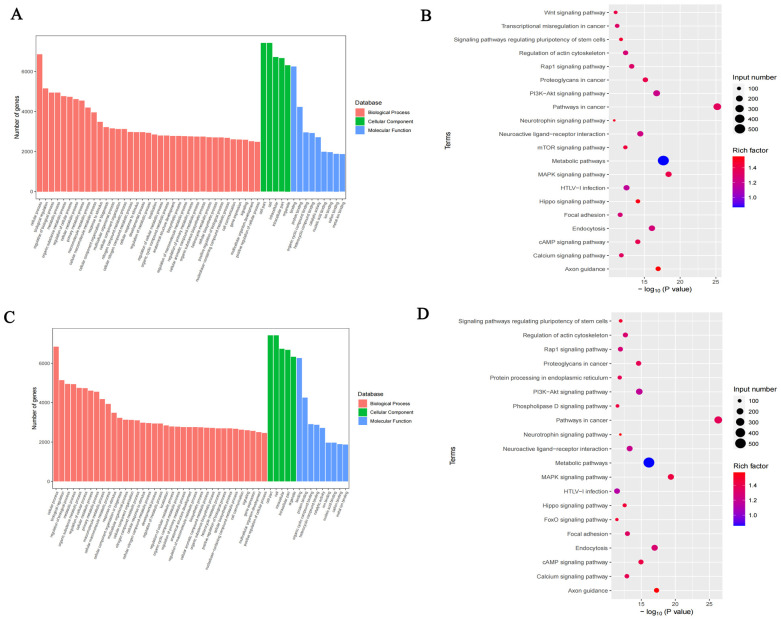
Pathway enrichment of differentially methylated m^6^A sites (DMMSs). (**A**). GO analysis of hypermethylated genes. (**B**). KEGG analysis of hypermethylated genes. (**C**). GO analysis of hypomethylated genes. (**D**). KEGG analysis of hypomethylated genes.

**Figure 8 ijms-25-01990-f008:**
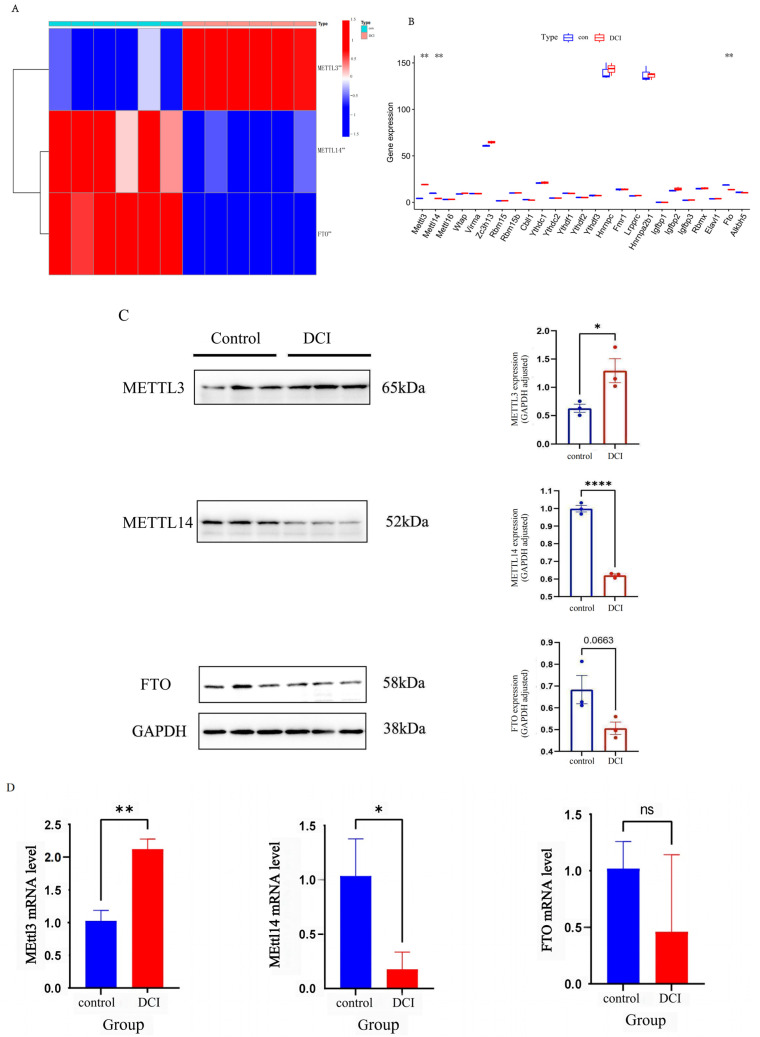
(**A**). Box line plot of RNA expression of 26 m^6^A RNA methylation regulators in the hippocampus of control (*n* = 3) and DCI (*n* = 3) group of mice. (**B**). Heat map of METTL3, METTL14, and FTO expression in the hippocampus of mice in control and DCI group. (**C**). Expressions of METTL3, METTL14, and FTO in the hippocampus were detected via Western blot using anti METTL3, METTL14, and FTO antibody, respectively GAPDH was used as loading control values are means ± SEM. (**D**). qRT-PCR of METTL3, METTL14, and FTO. Values are means ± SEM. *n* = 3. Significance identification: Not significant; ns, *p* ≥ 0.05; *, *p* < 0.05; **, *p* < 0.01; ****, *p* < 0.0001vs. control group.

**Figure 9 ijms-25-01990-f009:**
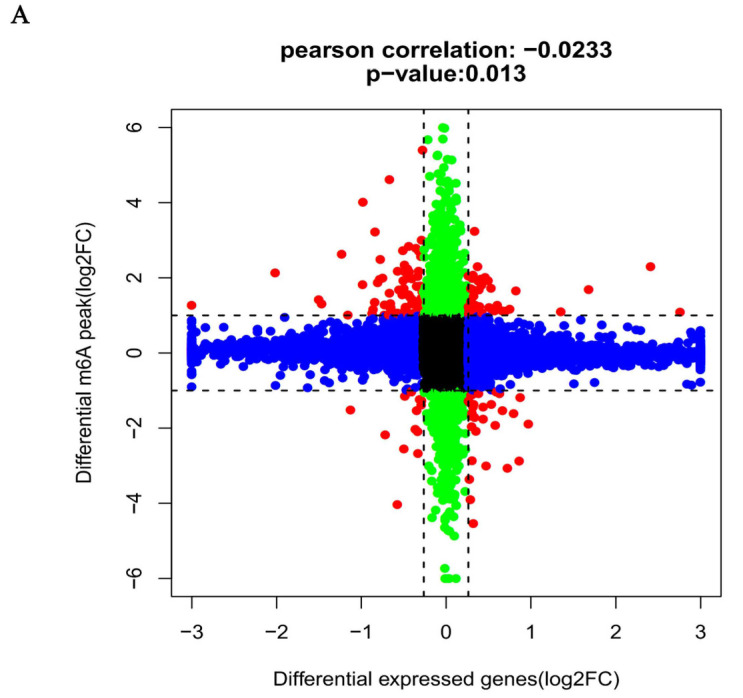
Joint analysis of m^6^A methylation and mRNA expression. (**A**). Nine quadrant graph diagram shows the relationship between mRNA m^6^A methylation and its mRNA expression. (**B**). The protein–protein interaction network shows the connection between the proteins of the genes used in the combined analysis. green: m^6^A upregulated and mRNA downregulated; red: m^6^A upregulated and mRNA upregulated; yellow: m^6^A downregulated and mRNA downregulated; blue: m^6^A downregulated and mRNA upregulated.

**Figure 10 ijms-25-01990-f010:**
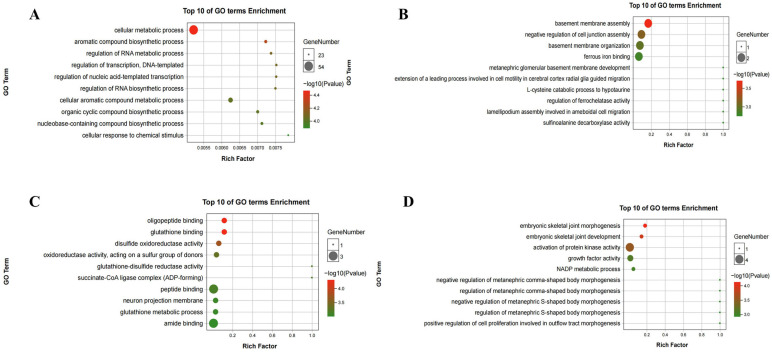
GO and KEGG analysis analyses of the 163 DEGs and DMMSs. (**A**). GO analysis of m^6^A upregulated and mRNA downregulated genes. (**B**). GO analysis of m^6^A upregulated and mRNA upregulated genes. (**C**). GO analysis of m^6^A downregulated and mRNA downregulated genes. (**D**). GO analysis of m6A downregulated and mRNA upregulated genes (**E**). KEGG analysis of m6A upregulated and mRNA downregulated genes. (**F**). KEGG analysis of m^6^A upregulated and mRNA upregulated genes. (**G**). KEGG analysis of m^6^A downregulated and mRNA downregulated genes. (**H**). KEGG analysis of m^6^A downregulated and mRNA upregulated genes.

**Table 1 ijms-25-01990-t001:** Top twenty differentially expressed genes in the hippocampus of DCI mice.

Gene	Description	Chromosome	log2FC	*p*-Value	Pattern
*Ndufa12*	NADH:ubiquinone oxidoreductase subunit A12 [Mus musculus (house mouse)]	10	21.70	0.034	up
*Klc1*	kinesin light chain 1 [Mus musculus (house mouse)]	12	18.19	0.020	up
*Map7d1*	MAP7 domain containing 1 [Mus musculus (house mouse)]	4	18.01	0.043	up
*Ccdc124*	coiled-coil domain containing 124 [Mus musculus (house mouse)]	8	15.51	0.017	up
*Gtf2f1*	general transcription factor IIF, polypeptide 1 [Mus musculus (house mouse)]	17	13.92	0.026	up
*Dbn1*	drebrin 1 [Mus musculus (house mouse)]	13	12.82	0.023	up
*Palm*	paralemmin [Mus musculus (house mouse)]	3	12.30	0.018	up
*Rrp1*	ribosomal RNA processing 1 [Mus musculus (house mouse)]	1	11.72	0.038	up
*Sfpq*	splicing factor proline/glutamine rich (polypyrimidine tract binding protein associated) [Mus musculus (house mouse)]	4	11.36	0.013	up
*Trir*	telomerase RNA component interacting RNase [Mus musculus (house mouse)]	8	9.92	0.015	up
*Gabarapl1*	GABA type A receptor associated protein like 1 [Mus musculus (house mouse)]	6	−21.73	0.001	down
*Eef1a1*	eukaryotic translation elongation factor 1 alpha 1 [Mus musculus (house mouse)]	9	−20.84	0.006	down
*Septin4*	septin 4 [Mus musculus (house mouse)]	11	−14.17	0.014	down
*Cox6b1*	cytochrome c oxidase, subunit 6B1 [Mus musculus (house mouse)]	7	−10.71	0.002	down
*Atf4*	activating transcription factor 4 [Mus musculus (house mouse)]	15	−10.46	0.023	down
*Nsg1*	neuron specific gene family member 1 [Mus musculus (house mouse)]	5	−9.15	0.008	down
*Fos*	FBJ osteosarcoma oncogene [Mus musculus (house mouse)]	12	−7.36	0.026	down
*Eif1*	eukaryotic translation initiation factor 1 [Mus musculus (house mouse)]	18	−7.32	0.034	down
*Tuba1b*	tubulin, alpha 1B [Mus musculus (house mouse)]	15	−7.21	0.007	down
*Mdh1*	malate dehydrogenase 1, NAD (soluble) [Mus musculus (house mouse)]	1	−7.13	0.043	down

**Table 2 ijms-25-01990-t002:** The top twenty differentially methylated peaks in the hippocampus in DCI mice.

Gene	Official Full Name	Chromosome	log2FC	*p*-Value	Peak Region	Pattern
*Foxb2*	forkhead box B2 [Mus musculus (house mouse)]	19	5.68	0.006	CDS	up
*Tex15*	testis expressed 15 [Mus musculus (house mouse)]	8	5.65	0.035	CDS	up
*Angpt2*	angiopoietin 2 [Mus musculus (house mouse)]	8	5.40	7.60 × 10^−5^	3′UTR	up
*Snai2*	snail family transcriptional repressor 2 [Mus musculus (house mouse)]	16	5.25	0.004	CDS	up
*Fstl5*	follistatin like 5 [Mus musculus (house mouse)]	3	5.06	2.73 × 10^−4^	3′UTR	up
*Tgfb2*	transforming growth factor beta 2 [Mus musculus (house mouse)]	1	5.01	0.007	CDS	up
*Lrrc2*	leucine rich repeat containing 2 [Mus musculus (house mouse)]	9	4.98	7.59 × 10^−9^	3′UTR	up
*Ccdc62*	coiled-coil domain containing 62 [Mus musculus (house mouse)]	5	4.98	0.001	3′UTR	up
*B3gnt3*	UDP-GlcNAc:betaGal beta-1 [Mus musculus (house mouse)]	8	4.94	0.006	3′UTR	up
*4930533K18Rik*	RIKEN cDNA 4930533K18 gene [Mus musculus (house mouse)]	10	4.77	0.014	3′UTR	up
*Igbp1b*	immunoglobulin (CD79A) binding protein 1b [Mus musculus (house mouse)]	6	−3.31	0.029	CDS	down
*Rassf10*	Ras association (RalGDS/AF-6) domain family (N-terminal) member 10 [Mus musculus (house mouse)]	7	−2.88	2.31 × 10^−5^	CDS	down
*Rhoh*	ras homolog family member H [Mus musculus (house mouse)]	5	−2.59	0.037	3′UTR	down
*N4bp2*	NEDD4 binding protein 2 [Mus musculus (house mouse)]	5	−2.48	0.006	CDS	down
*Txndc2*	thioredoxin domain containing 2 (spermatozoa) [Mus musculus (house mouse)]	17	−2.46	0.020	5′UTR	down
*Rab7b*	RAB7B, member RAS oncogene family [Mus musculus (house mouse)]	1	−2.40	0.039	3′UTR	down
*Gm5093*	predicted gene 5093 [Mus musculus (house mouse)]	17	−2.40	4.31 × 10^−5^	CDS	down
*Hist2h3c2*	H3 clustered histone 15 [Mus musculus (house mouse)]	3	−2.27	0.002	3′UTR	down
*P2ry1*	purinergic receptor P2Y [Mus musculus (house mouse)]	3	−2.24	0.039	CDS	down
*Erich5*	glutamate rich 5 [Mus musculus (house mouse)]	15	−2.17	0.013	CDS	down

## Data Availability

The data that support the findings of this study are available from the corresponding author upon request; please contact the corresponding author via email.

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
