# Peer review of "Altered N6-Methyladenosine Modification Patterns and Transcript Profiles Contributes to Cognitive Dysfunction in High-Fat Induced Diabetic Mice"

_ijms, 2024, doi:10.3390/ijms25041990_

Round 1

Reviewer 1 Report

Comments and Suggestions for Authors

My specific comments are listed below:

-       The graphic abstract is not very readable

-       Line 26: The headline should not be left alone at the bottom of the page

-       Line 28: lack of spacing

-       Paragraph 1: how MCI is connected to DCI? Does MCI always develop in DCI? Is DCI a special case of MCI? How often does DCI accompany T2DM?

-       Line 52: double bracket

-       Line 60: no explanation of abbreviations

-       Line 77: The headline should not be left alone at the bottom of the page

-       Line 78: some section titles are neutral (e.g.:2.2,2.3), while others indicate what was observed (e.g.: 2.1, 2.5)- please unify them

-       Line 86: higher?

-       Lines 85-87: “The fasting blood glucose concentration was significantly  high in both the groups …. than in the control group” - unclear sentence

-       Figure 1A - unit is missing

-       Figure 1B - double bracket 

-       Figure  1B - Why the sudden increase in week 5?

-       Figure 1 - What does the graph mean? median? SD? SEM?

-       Figure 1 - what is the "n" number?

-       Line 95: “shorter” on what day?

-       Line 96-98: “The difference was not statistically significant (p = 0.985), but the escape latency 96 of mice in the control group was significantly shorter than that of mice in the DCI group 97 on days 2–4 (Figure 2A).” - unclear sentence

-       Line 100: only on day 5? on others not? why? day 5 of what? which week of the experiment?

-       Figure 2 D/E: Are these the only tests to confirm DCI?

-       Figure 2B/C/D/E: On what day/week of the experiment were the tests performed?

-       Figure 2A: what's on the X-axis?

-       Figure 2 B/C/D/E: What represents boxes and whiskers on the box and whisker plots? what is the "n" number? individual observations in the same colour as the box are not visible, no explanation for *,**,***, ns

-       Line 111: different font

-       Figure 3: unreadable scale, Based on what criteria were 3 photos selected from 9 individuals? how many stains per individual were taken

-       Figure 4: figure 4 is mentioned in the text earlier so it should appear before Table 1

-       Table 1: The full names of the genes or the category to which they belong should be given

-       Figure 4 D&E: why the same categories are presented as “the most enriched GO analysis of down-regulated genes” and “ the most enriched pathway analysis of up-regulated genes”

-       Figure 4: Where did such categories come from? Were the annotations when analysing the database verified? Why when analysing the genes in cerebral cortex categories like “colorectal cancer” and “endometrial cancer” appear? 

-       Line 145- 146: “ In contrast, 23,751 non-overlapping m6A peaks in the DCI group were found within 10,809 mRNAs in the three replicates of the control group.” - unclear sentence

-       Line 148: no explanation of abbreviations

-       Figure 5: figure 5 is mentioned in the text earlier so it should appear before Table 2

-       Line 155-156: Figure 6b lacks an indication of statistical significance, “enriched” in relation to what?

-       Table 2: Table 1: The full names of the genes or the category to which they belong should be given

-       Figure 5: poor quality – unreadable

-       Figure 6: too small not readable

-       Line 169: space is missing

-       figure 7: what's on the Y axis? No changes on the Y chromosome? what's on the chart? The mean? The median? Why are there no deviations? significance? from how many individuals is the data. what group does the graph show?

-       line 186: please explain exactly how the gene from the hippocampus fell into the category “small cell lung cancer”?

-       figure 8A/C: too small not readable, 

-       figure 8: what group does the graph show?

-       Line 206: 9A?9b?9c?9d? all of them?

-       Line 208: Why  alkbh5 hasn't it been analyzed?

-       Line 213: With exactly which sequencing results? Please indicate the figure  or paragraph

-       Figure 9 A/B: too small not readable,

-       Figure 9: what is the "n" number? Protein level, gene expression. Why hippocampus is in the section titled “…expression of genes in the cerebral cortex result….”?

-       Figure 9C: FTO - why not label with symbols?

-       Line 216: “in the brain” – in the hippocampus? In the cortex? In the whole brain?

-       Section 2.7 - For which experimental group? For what part of the brain?

-       Line 243 - he established abbreviations should be used throughout the text

-       Figure 10 B: too small not readable

-       Figure 11 E/F/G/H: too small not readable

-       Line 421: for how many animals the staining was done?

-       Line 426: please describe in brief

-       Line 429: how many animals?

-       Line 458: DEG & DMMS - abbreviations should appear earlier in the text

-       Section 4: no description of the western blot procedure

-       Section 4.9: how the quality and quantity of RNA were checked?

-       Line 504: “as mean ± standard deviation (SD) for continuous numerical data with 504 normal distribution” - Which analyzed features met these criteria? and how were the others shown?

-       Line 505: How was the normality of the distribution checked?

-       No reference in the text to supplementary Table 1 

-       double numbering in references

Author Response

Comments

  1. Summary

Thank you very much for your careful review and valuable comments on our submitted manuscript. We have revised and improved the manuscript according to your suggestions, and we would like to show you our responses and revisions.

2.Questions for General Evaluation

Reviewers Evaluation

Response and Revisions

Does the introduction provide sufficient background and include all relevant references?

Can be improved

I have give my corresponding response in the point-by-point response letter

Are all the cited references relevant to the research?

Can be improved

Is the research design appropriate?

Can be improved

Are the methods adequately described?

Must be improved

Are the results clearly presented?

Must be improved

Are the conclusions supported by the results?

Can be improved

  1. Point-by-point response to Comments and Suggestions for Authors

Comments 1:The graphic abstract is not very readable

Response 1:Dear reviewer, after seeing your comment all of us authors reviewed the graphical abstract again, given that the graphical abstract exists for a reason: to explain the information of the research paper in a clear and attractive way. Our graphic abstract shows the steps we took in this research and correspondingly presents the results of each part of the research. Therefore, we believe that our graphic abstract meets the requirements of the journal.

Comments 2:Line 26: The headline should not be left alone at the bottom of the page

Response 2: Thank you for pointing this out ,we read the journal's layout requirements again, reformatted the article, and the title is no longer left alone at the bottom of our resubmitted manuscript.

Comments 3:Line 28: lack of spacing

Response 3:Dear reviewer, As this journal is provided with a template, the line spacing of our text is consistent with the template provided by the journal.

-Comments 4:Paragraph 1: how MCI is connected to DCI? Does MCI always develop in DCI? Is DCI a special case of MCI? How often does DCI accompany T2DM?

Response 4:We would be happy to explain the definition of DCI and its relation to MCI to help you better understand our article: DCI usually refers to cognitive impairment in people with diabetes mellitus, a decline in cognitive functioning that occurs earlier compared to peers. So you can think of diabetes-induced cognitive impairment as including mild cognitive dysfunction marked by memory loss (which is what this article is all about). 2021 American Diabetes Association (ADA) guidelines make clear reference to the importance of recognising cognitive impairment in diabetes, stating that poor glycaemic control is associated with cognitive decline, and that cognitive functioning is worse the longer the duration of the diabetes disease. 2 The risk of Alzheimer's disease and vascular dementia in people with type 2 diabetes is 1.46 and 2.48 times higher than in non-diabetics, respectively.

The onset of DCI is often very insidious. In the early stage of diabetic cognitive dysfunction, there may only be a decline in one or more of the six abilities: learning and memory, language, executive function, sensorimotor function, complex attention and social cognition, which will not affect life and work, and therefore often does not attract people's attention. However, as the disease progresses, mental, behavioural and personality abnormalities may appear and lead to a syndrome in which the patient's ability in daily life, learning, work and social interaction is significantly reduced.

Comments 5:Line 52: double bracket

Response 5:Thank you for pointing this out, We removed the redundant brackets

Comments 6:Line 60: no explanation of abbreviations

Response 6:Thank you for pointing this out, After discussion among all our authors, it was decided to remove the acronym "IEGs", which will not be used in the rest of the text.

Comments 7:Line 77: The headline should not be left alone at the bottom of the page

Response 7:Thank you for pointing this out, we read the journal's layout requirements again, reformatted the article, and the title is no longer left alone at the bottom of our resubmitted manuscript.

Comments 8:Line 78: some section titles are neutral (e.g.:2.2,2.3), while others indicate what was observed (e.g.: 2.1, 2.5)- please unify them

Response 8:We re-examined the subheadings of the Result module and compared them to articles of the same type as ours. The paragraph headings of each paragraph exist in the sense that they guide the reader to a better reading of the article and allow the reader to quickly locate the required information. Our paragraph titles fulfil this purpose. Section titles can be either descriptive (what you call neutral) or explanatory (what you call observations), and the journals we submit to do not have writing requirements that require authors to align all subheadings descriptively or explanatorily.

Comments 9:Line 86: higher?

Response 9:To make it clearer what we want to say, we have rewritten the sentence in lines 85-87 (currently in lines 88-91, which have been highlighted in dark blue)

Comments 10:Lines 85-87: “The fasting blood glucose concentration was significantly high in both the groups …. than in the control group” - unclear sentence

Response 10:We have rewritten the sentence to avoid ambiguity:One week after the start of the experiment, both groups had elevated fasting blood glucose, and it was significantly higher in the DCI group than in the control group

Comments 11:Figure 1A - unit is missing

Response 12:Changes have been made to the picture

Comments 12:Figure 1B - double bracket

Response 12:Changes have been made to the picture

Comments 13:Figure  1B - Why the sudden increase in week 5?

Response 13:It is important that we present the results of our observations honestly in our article, all we can say for sure is that we have been using the same instrument and the same two researchers have been measuring the blood glucose (one researcher was recording while the other was measuring) so we can guarantee that the reason for this result is not due to a technical error at all. And we did not change the diet during the experiment. We are still trying to find out why there was a sudden increase in blood glucose at week 5, but we can assure you that the results we present are correct. But we can confirm that the results we present are true.

Comments 14:Figure 1 - What does the graph mean? median? SD? SEM?

Response 14:Mean ± SD

Comments 15:Figure 1 - what is the "n" number?

Response 15:9 mice per group (groupings and related information are available in 4. Materials and Methods 4.1. Animals, already highlighted)

Comments 16:Line 95: “shorter” on what day?

Response 16:We have rewritten the sentence for easier understanding:all mice showed shorter escape latency on day 5 than on day 1.

Comments 17:Line 96-98: “The difference was not statistically significant (p = 0.985), but the escape latency 96 of mice in the control group was significantly shorter than that of mice in the DCI group 97 on days 2–4 (Figure 2A).” - unclear sentence

Response 17:We have rewritten the sentence for easier understanding:“ The difference was not statistically significant on day 1(p = 0.985), but the escape latency of mice in the control group was significantly shorter than that of mice in the DCI group on days 2–4 (Figure 2A).

Comments 18:Line 100: only on day 5? on others not? why? day 5 of what? which week of the experiment?

Response 18:The aim of our Morris water maze experiment was to observe the success of the animal model, so this experiment was performed a total of 5 times comparing the last and first swimming distances, and was included in subsequent experiments when the DCI model of the animal was found to be created successfully. This experiment was conducted in week 7 (started immediately after the successful creation of the T2DM model)

Comments 19:Figure 2 D/E: Are these the only tests to confirm DCI?

Response 19:The Morris water maze experiment mainly measures spatial learning and memory ability, and the main experiments include the hidden station test, spatial exploration test, reverse test and visual station test. We determined that after establishing the T2DM model mice, the success of DCI modelling can be determined by observing the changes in cognitive ability and spatial orientation of the mice. This is the classic method for determining the success of DCI modelling, and in order to comply with the 3Rs (avoid unnecessary injury, starvation, discomfort, fright, torture, illness and pain, and ensure that natural behaviours can be maximised). We minimise the number of experiments carried out on animals while ensuring that the results are correct.

Comments 20:Figure 2B/C/D/E: On what day/week of the experiment were the tests performed?

Response 20:After establishing the success of the T2DM animal model, we immediately carried out the water maze experiment (week 7, 5 days).

Comments 21:Figure 2A: what's on the X-axis?

Response 21:Changes were made to the figure

Comments 22:Figure 2 B/C/D/E: What represents boxes and whiskers on the box and whisker plots? what is the "n" number? individual observations in the same colour as the box are not visible, no explanation for *,**,***, ns

Response 22:Changes were made to the figure

Comments 23:Line 111: different font

Response 23:Thank you for pointing this out, We have standardised the font throughout the text (font: times new roman; font size: 10).

Comments 24:Figure 3: unreadable scale, Based on what criteria were 3 photos selected from 9 individuals? how many stains per individual were taken

Response 24:We randomly selected 3 mice out of 9 mice in each group for H&E staining. Staining was performed once per section. Also our vector images have been packaged and uploaded.

Comments 25:Figure 4: figure 4 is mentioned in the text earlier so it should appear before Table 1

Response 25: Thank you for pointing this out, Figure 4 and table 1 have been switched.

Comments 26:Table 1: The full names of the genes or the category to which they belong should be given

Response 26:I have added a column to Tables 1 and 2 to describe this information

Comments 27:Figure 4 D&E: why the same categories are presented as “the most enriched GO analysis of down-regulated genes” and “ the most enriched pathway analysis of up-regulated genes”

Response 27: The meaning of D and E in Figure 4 is as follows::D)The enriched KEGG enrichment analysis of up-regulated genes.. E)The enriched KEGG enrichment analysis of down-regulated genes.D and E are both KEGG enrichment analyses, and B & C in Figure 4 are GO analyses.

Comments 28:Figure 4: Where did such categories come from? Were the annotations when analysing the database verified? Why when analysing the genes in cerebral cortex categories like “colorectal cancer” and “endometrial cancer” appear?

Response 28:The region of focus of this study is the hippocampus (we have made a unification of the full text), and in response to your query we have answered the following: 1) A cell contains all the genetic information of a species 2) The pleiotropy of genes. Therefore colorectal and endometrial cancers will appear when we do enrichment analysis of differentially expressed genes. I would like to give a related article on brain trauma for your reference: Yu J, Zhang Y, Ma H, Zeng R, Liu R, Wang P, Jin X, Zhao Y. Epitranscriptomic profiling of N6-methyladenosine-related RNA methylation in rat cerebral cortex following traumatic brain injury. Mol Brain. 2020 Jan 28;13(1):11. doi: 10.1186/s13041-020-0554-0. PMID. 31992337; PMCID: PMC6986156.Also for database validation we are willing to give you the following URLs: http://go-database-sql.org/; https://www.kegg.jp/.

Comments 29:Line 145- 146: “ In contrast, 23,751 non-overlapping m6A peaks in the DCI group were found within 10,809 mRNAs in the three replicates of the control group.” - unclear sentence

Response 29:We have changed the sentences to make the article better understood:We have changed the sentences to make the article better understood:In contrast, the DCI group exhibited 23,751 non-overlapping m6A peaks across 10,809 distinct mRNAs in the three replicates of the control group. Three replicates: additionally in our experimental design it was ensured that there were at least three replicates of each experiment (three mice)

Comments 30:Line 148: no explanation of abbreviations

Response 30:Replenished (dark blue)

Comments 31:Figure 5: figure 5 is mentioned in the text earlier so it should appear before Table 2

Response 31:The positions of figure 5 and table 2 have been switched.

Comments 32:Line 155-156: Figure 6b lacks an indication of statistical significance, “enriched” in relation to what?

Response 32:The description of our original article is as follows:”Distribution of Peak in the different areas on gene exons. The horizontal coordinates are the 5'UTR, CDS, and 3'UTR,and the vertical coordinates are the relative proportions of the distribution of different peak regions. ”.There is a peak in this graph, and the significance of this peak is that the m6A modification is enriched in the 3' UTR near the mRNA termination codon.

Comments 33:Table 2: Table 1: The full names of the genes or the category to which they belong should be given

Response 33:has been added

Comments 34:Figure 5: poor quality – unreadable

Response 34:Have replaced the image and uploaded the vector figure

Comments 35:Figure 6: too small not readable

Response 35:Have replaced the image and uploaded the vector figure

Comments 36:Line 169: space is missing

Response 36:Thank you for pointing this out,has been added

Comments 37:figure 7: what's on the Y axis? No changes on the Y chromosome? what's on the chart? The mean? The median? Why are there no deviations? significance? from how many individuals is the data. what group does the graph show?

Response 37:We plotted Figure 7 originally to give the reader a visualisation of the distribution of differentially expressed genes on the chromosomes, but as our subsequent analysis is not relevant, we will not describe it again in order to avoid any misunderstanding on the part of the reader.

Comments 38:line 186: please explain exactly how the gene from the hippocampus fell into the category “small cell lung cancer”?

Response 38:(i) One cell contains all the genetic information of the species (ii) Multiplicity of genes. The purpose of doing KEGG analysis in this study is to find the genes that are differentially expressed in the two groups enriched in those pathways. Small cell lung cancer is not a trait studied in this paper and does not need to be explained. Reference:Yu J, Zhang Y, Ma H, Zeng R, Liu R, Wang P, Jin X, Zhao Y. Epitranscriptomic profiling of N6-methyladenosine-related RNA methylation in rat cerebral cortex following traumatic brain injury. Mol Brain. 2020 Jan 28;13(1):11. doi: 10.1186/s13041-020-0554-0. PMID: 31992337; PMCID: PMC6986156.

Comments 39:figure 8A/C: too small not readable,

Response 39:Replaced the figure and uploaded the vector figure

Comments 40:figure 8: what group does the graph show?

Response 40:Figure 8. Pathway enrichment of Differentially methylated m6A sites (DMMSs). A. GO analysis of hypermethylated genes. B. KEGG analysis of hypermethylated genes. C. GO analysis of hypomethylated genes. D. KEGG analysis of hypomethylated genes.

Comments 41:Line 206: 9A?9b?9c?9d? all of them?

Response 41:Figure 9B(The original text has been changed.)

Comments 42:Line 208: Why  alkbh5 hasn't it been analyzed?

Response 42:Alkbh5 expression in this experiment did not differ between the two groups.

Comments 43:Line 213: With exactly which sequencing results? Please indicate the figure  or paragraph

Response 43:already specified

Comments 44:Figure 9 A/B: too small not readable,

Response 44:Replaced the figure and uploaded the vector figure

Comments 45:Figure 9: what is the "n" number? Protein level, gene expression. Why hippocampus is in the section titled “…expression of genes in the cerebral cortex result….”?

Response 45:"n" number has been added; images have been replaced; full text has been harmonised

Comments 46:Figure 9C: FTO - why not label with symbols?

Response 46:Figure have been replaced

Comments 47:Line 216: “in the brain” – in the hippocampus? In the cortex? In the whole brain?

Response 47: Our study focused on hippocampal, which has been standardised throughout.

Comments 48:Section 2.7 - For which experimental group? For what part of the brain?

Response 48:Experimental group: DCI group, hippocampal region

Comments 49:Line 243 - he established abbreviations should be used throughout the text

Response 49:Abbreviations have been standardised throughout the text

Comments 50:Figure 10 B: too small not readable

Response 50:Replaced the figure and uploaded the vector figure

Comments 51:Figure 11 E/F/G/H: too small not readable

Response 51:Replaced the figure and uploaded the vector figure

Comments 52:Line 421: for how many animals the staining was done?

Response 52:Researchers randomly selected three mice from each of the two groups for Haematoxylin and eosin (H&E) staining

Comments 53:Line 426: please describe in brief

Response 53:Have been added to the manuscript

Comments 54:Line 429: how many animals?

Response 54:Have been added to the manuscript

Comments 55:Line 458: DEG & DMMS - abbreviations should appear earlier in the text

Response 55:Changes have been made to the manuscript

Comments 56:Section 4: no description of the western blot procedure

Response 56:The WB procedure is described in "4.9. Validation of gene expression levels".

Comments 57:Section 4.9: how the quality and quantity of RNA were checked?

Response 57:The quantification and assessment of RNA concentration and purity through UV spectrophotometry.

Comments 58:Line 504: “as mean ± standard deviation (SD) for continuous numerical data with 504 normal distribution” - Which analyzed features met these criteria? and how were the others shown?

Response 58:We need to test the normality and variance chi-square of the distribution before performing the analysis of differences in the qRT-PCR results, we used the shapiro-wilk test.The R language was used and the code was as follows:

exp <- read.csv("H:/R_project/qPCR/exp.csv")

shapiro.test(exp[which(exp$Group=="control"),"value"])

shapiro.test(exp[which(exp$Group=="treatment"),"value"])

Comments 59:Line 505: How was the normality of the distribution checked?

Response 59:Normal distribution test using the Shapiro-Wilk test. Homogeneity of variance test using the Bartlett test.

Comments 60:No reference in the text to supplementary Table 1

Response 60:Thank you for pointing this out, we have added this sentence to section 4.9:Primers used in this study shown in Table S1

Comments 61:double numbering in references

Response 61:Auto-numbering in journal templates has been removed

Reviewer 2 Report

Comments and Suggestions for Authors

In this manuscript, Cao and colleagues studied the alteration of m6A modification in cognitive dysfunction in high-fat induced diabetic mice. They utilized mouse DCI model and performed RNA-seq and MeRIP-seq. The concepts in this manuscript is novel. However, a number of caveats needs to be addressed.

1.       It is well known that m6A mRNA modification play key roles in several human diseases. First, the m6A-seq are not correctly interpreted. In mammalian cells, only 10% or less m6A modification, is in mRNA. From the method part, no poly-A+ purification was performed. If the authors intend to suggest the alteration of m6A levels on mRNA, m6A-seq should performed on purified cellular mRNA. Thus, two rounds of poly-A+ selection need to be performed before the MeRIP-seq assay.

2.       The author attempts to study the changes of m6A in DCI, but there is a clear and significant issue that remains unresolved: whether DCI truly affects M6A levels? This can be directly verified through experiments, such as using hippocampal tissue for LC/MS and mRNA m6A dot blot assay.

3.       Please check the figure legends and annotate the ‘n number’ for each experiment.

4.       Please briefly explain the reasons for using only male mice.

Author Response

Comments

  1. Summary

Thank you very much for your careful review and valuable comments on our submitted manuscript. We have revised and improved the manuscript according to your suggestions, and we would like to show you our responses and revisions.

2.Questions for General Evaluation

Reviewers Evaluation

Response and Revisions

Does the introduction provide sufficient background and include all relevant references?

Can be improved

I have give my corresponding response in the point-by-point response letter

Are all the cited references relevant to the research?

Yes

Is the research design appropriate?

Can be improved

Are the methods adequately described?

Must be improved

Are the results clearly presented?

Can be improved

Are the conclusions supported by the results?

Can be improved

  1. Point-by-point response to Comments and Suggestions for Authors

Comments 1:It is well known that m6A mRNA modification play key roles in several human diseases. First, the m6A-seq are not correctly interpreted. In mammalian cells, only 10% or less m6A modification, is in mRNA. From the method part, no poly-A+ purification was performed. If the authors intend to suggest the alteration of m6A levels on mRNA, m6A-seq should performed on purified cellular mRNA. Thus, two rounds of poly-A+ selection need to be performed before the MeRIP-seq assay.

Response 1: More than 100 different types of post-transcriptional modifications have been identified on RNAs, and all known RNA species can be modified, with rRNAs and tRNAs being the most modified. mRNAs are mainly modified with m6A, m1A and m5C. m6A is the most abundant mRNA modification, and is also the most well-studied modification of RNA. m6A modifications are present in mRNAs, lincRNAs, pri-miRNAs and rRNAs. The experimental principle of meRIP-seq is shown below: firstly, mRNA is captured by polyA, or rRNA is removed by rRNA knockout, and then the obtained RNA is broken into small fragments of about 100 nt; after that, an antibody specific for a particular modification (m6A antibody for example) is used for After that, the m6A-modified RNA fragments were enriched and recovered by immunoprecipitation using a specific modification-specific antibody (m6A antibody, for example); the recovered RNA was then subjected to library construction, high-throughput sequencing and bioinformatics analysis, and the m6A-modified sites were identified by peak analysis.

Since rRNA contains m6A modification, rRNA rejection during the experiment is necessary. Currently, there are two types of techniques to remove rRNA interference, one is polyA capture of positively sieved mRNA, and the other is rRNA rejection of negatively sieved rRNA, which has a huge difference in cost, as well as significant differences in their detection targets and results:

-mRNA capture: only m6A modifications on mRNAs and some IncRNAs with polyA tails can be investigated, which is the most commonly used method at present;

-rRNA knockout: limited by species, but the method is very comprehensive in detecting m6A modifications on mRNAs, IncRNAs, circRNAs and pre-mRNAs;

In this study, polyA capture of positively screened mRNA was used.

Comments 2:The author attempts to study the changes of m6A in DCI, but there is a clear and significant issue that remains unresolved: whether DCI truly affects M6A levels? This can be directly verified through experiments, such as using hippocampal tissue for LC/MS and mRNA m6A dot blot assay.

Response 2:Thanks to your suggestion, we already know that neural tissues have higher levels of RNA methylation than other tissues (Meyer KD, Saletore Y, Zumbo P, Elemento O, Mason CE, Jaffrey SR. Comprehensive analysis of mRNA methylation reveals enrichment in 3' UTRs and near stop codons. Cell. 2012 Jun 22;149(7):1635-46. doi: 10.1016/j.cell.2012.05.003. Epub 2012 May 17. PMID. 22608085; PMCID: PMC3383396.) m6A can indeed affect cognitive functions in organisms, and its related mechanisms are being explored by an increasing number of researchers (Chokkalla AK, Mehta SL, Vemuganti R. Epitranscriptomic regulation by m6A RNA methylation in brain development and diseases. J Cereb Blood Flow Metab. 2020 Dec;40(12):2331-2349. doi: 10.1177/0271678X20960033. Epub 2020 (Sep 23. PMID: 32967524; PMCID: PMC7820693.). Given that DCI can be understood as MCI due to diabetes (Luo A, Xie Z, Wang Y, Wang X, Li S, Yan J, Zhan G, Zhou Z, Zhao Y, Li S. Type 2 diabetes mellitus-associated cognitive dysfunction. Advances in potential mechanisms and therapies. Neurosci Biobehav Rev. 2022 Jun;137:104642. doi: 10.1016/j.neubiorev.2022.104642. Epub 2022 Mar 30 . PMID: 35367221.), a large number of previous experiments have confirmed that MCI as well as AD lead to changes in m6A, and therefore will not be repeated in this study!

The purpose of this study was to explore the gene expression status and m6A modification level in the hippocampus of mice with diabetes-induced cognitive impairment to pave the way for the subsequent exploration of the mechanism of the development of DCI (which is described in the Discussion), that is to say, in this study, we paid more attention to the question of whether the changes were made, how they were made, which m6A-regulated genes were altered, and in which pathways these changes were enriched. enriched in which pathways.

Comments 3:Please check the figure legends and annotate the ‘n number’ for each experiment.

Response 3:Thank you for pointing this out, we've optimised the image slices and labelled the number of animals per experiment.

Comments 4:Please briefly explain the reasons for using only male mice.

Response 4: Thank you for pointing this out, “In order to mitigate the potential influence of estrogen and the physiological cycle on the outcomes of the experiment, exclusively male mice were used.

Round 2

Reviewer 2 Report

Comments and Suggestions for Authors

I have no further concerns.